# Assessment of Color Perception and Preference with Eye-Tracking Analysis in a Dental Treatment Environment

**DOI:** 10.3390/ijerph18157981

**Published:** 2021-07-28

**Authors:** Eun-Sung Song, Won-Hyeon Kim, Beom-Hui Lee, Dong-Wook Han, Jong-Ho Lee, Bongju Kim

**Affiliations:** 1School of Integrated Technology, IIT (Institute of Integrated Technology), Gwangju Institute of Science and Technology (GIST), Gwangju 61005, Korea; eunsung@gist.ac.kr; 2Dental Life Science Research Institute/Innovation Research & Support Center for Dental Science, Seoul National University Dental Hospital, Seoul 03080, Korea; wonhyun79@gmail.com; 3Research Center for Bio-Based Chemistry, Korea Research Institute of Chemical Technology, Ulsan 44429, Korea; dlqja111@naver.com; 4Department of Cogno-Mechatronics Engineering, College of Nanoscience & Nanotechnology, Pusan National University, Busan 46241, Korea; 5Department of Oral and Maxillofacial Surgery, School of Dentistry, Seoul National University, Seoul 03080, Korea; leejongh@snu.ac.kr

**Keywords:** color, psychological color, eye tracking, perception, preference, dental treatment environment

## Abstract

Nowadays, medical facilities are developing their treatment environment to provide better services to their patients. In particular, dental hospitals have been considered uncomfortable and uninviting spaces, which needs to change so that people can visit easily and feel more relaxed. However, only a few systematic studies have reported on the demand for building a comfortable space. This study aimed to investigate gaze characteristics based on a color preference survey of the dental unit chair, which has the most influence on spatial perception in the dental treatment environment, using an eye tracking technique for color. The results of this study showed that the color perception by eye tracking and the color preference by survey did not tend to match. The color most viewed by a majority of subjects was pink, which attracted a high level of attention, regardless of personal preference. In addition, for the psychological color images associated with color preference, the subjects tended to prefer images such as warmth, friendliness, and calmness. This appeared to reflect the psychology of the subjects who wished to replace their feelings of anxiety or fear when going to the dental hospital with comfort and tranquility. Therefore, colors that can provide comfort and tranquility to patients should be considered first as visual elements (e.g., brown) in creating a dental treatment environment.

## 1. Introduction

Recently, medical facilities have become interested in creating an environment that provides psychological stability and satisfaction to patients and healthcare workers through improvement of the treatment environment [1]. In particular, creating an environment preferred by patients can effectively relieve the anxiety of patients and promote their rehabilitation [2]. The indoor environment, including safety, lighting, sound, color, creative artworks, and furniture is an essential element to consider when constructing a medical environment [3,4]. Additionally, nature, neutral colors, the sound of treatment at the clinic, and cooperation with the patient’s family have been reported as factors that increase the treatment effect in the treatment environment [5]. It was recently reported that the use of a warm color (yellow) and a cool color (blue) together in a pediatric dentistry environment can create a positive treatment environment and that black and red have a negative effect [6]. In establishing such a treatment environment, the interaction of the medical environment and medical service provider with the patient is a very important factor in maintaining health care as patients do not visit places where they have experienced discomfort [7]. Therefore, the establishment of an ideal treatment environment effectively induces the recovery of visiting patients and improves the quality of medical services through indirect improvement of the doctor-patient relationship [7].

Many patients feel uncomfortable and anxious about pain and hospitals when they visit the dentist [8,9,10,11]. This suggests that hospital workers, including the doctors, need to create a psychologically satisfying environment when designing the dental space, not only for patients, but also for themselves. It helps to provide relaxed and comfortable spaces for patients that are designed using the rules of light and color in interior/exterior spaces [12,13]. The dental unit chair, where treatment is given directly to the patients, is one of the most important elements and its color design can provide a strong visual expression of the dental care space [14,15].

Therefore, this study aimed to perform gaze information analysis on dental unit chairs and color preference while surveying the images associated with color preference, and investigating the effect of color on emotions in order to prepare a plan for a systematic and rational color scheme. The degree of gaze on a specific space or area, information acquisition of the user over time, and gaze characteristics were examined based on the analysis of data obtained by eye tracking analysis. In order to create an aesthetically satisfying environment, color preference and color images were investigated through a survey, and the effect of colors in regard to color preference and emotions was examined. The survey regarding images that come to mind for color preference was conducted with the intention of understanding the effect of color on emotions to identify the colors that give users aesthetic satisfaction and psychological stability in a medical environment. For psychological color image analysis, the Color Image Scale is widely used as an emotional measure for color [16]. The image scale is the most basic tool for emotional color matching, color scheme or design, and increases objectivity and accuracy with respect to color. However, the image scale appears differently depending on the individual, country, culture, style, and environment. Therefore, an image space that is suitable for the characteristics of the subject should be created and used. In South Korea, the Research Institute for the Visual Language of Korea (Image Resource Institute) developed the interpersonal reactivity index (IRI) adjective image scale using the Kobayashi and IRI scales [17].

Studies to find the correlation between color and human behavior have reported that the information given by color is closely associated with emotion; these studies include analysis of people’s preference for color, the effect of color on emotions, and behavior and response to color [18,19,20,21,22,23,24]. However, there have not been any studies on emotional responses in a medical environment using color information. Accordingly, this study intended to ultimately present the color factors to consider in constructing an effective dental treatment environment, and identified ways to contribute to patients’ aesthetic satisfaction and psychological stability based on the color preference for the dental unit chair.

## 2. Materials and Methods

### 2.1. Eye Tracking

#### 2.1.1. Eye Tracking Apparatus and Subjects

In this study, gaze characteristics for 12 colors for the dental unit chair were acquired using an eye tracking apparatus. The experimental subjects included 22 men and women aged in their twenties to their forties. Each experiment was conducted individually for each subject for 60 s. During the experiment, the eye movements and fixed positions were monitored and recorded using a 15-inch laptop and an eye tracking apparatus (Tobii pro Nano, Tobii Technology Inc., Danderyd, Sweden) with a camera (Figure 1).

#### 2.1.2. Stimuli and Data Collection

The object used for data collection in this study was the dental unit chair, an important object in the dental treatment environment, and the object was expressed in 12 different colors as shown in Figure 2.

The color selection and eye movement were measured using 12 colors for the dental unit chair. After each stimulus was displayed for 5 s, a mask popped up for 5 s to avoid carryover and after-image effects. In order to balance the layout arrangement after masking, random arrangement was applied to prevent the same color from appearing in a fixed position. During the masking process, an X was displayed in the center of the screen to guide the eye movement back to the center, allowing the subject to start over from the center, away from the influence of the previous object. The eye movements were continuously monitored for a total of 60 s of image observation on 12 displays and the data were recorded 10 times.

#### 2.1.3. Procedure

Each subject participated in the experiment after the eye tracking apparatus was calibrated (Figure 3). After calibration, it was set up to display each frame. The experiment was conducted without prior explanation, and after the experiment, a survey and analysis of adjective images using the Kobayashi and IRI scales [19,20] were performed. Intrinsic color preference, color preference for the dental unit chair, and sensitivity to each image were examined.

### 2.2. Analysis of Eye Tracking Movement

#### 2.2.1. Setting the Visual Attention Time

Tobii, the equipment used in this experiment, recorded eye movements about 60 times per second (0.016 s) and measured the time the eyes were fixed at a specific area. A total of 3600 pieces of data were acquired with the measurement time set to 60 s. The spatial stimulus elements were determined based on the user gaze information by setting the time range for analysis and specifying the gaze characteristics according to the passage of time. However, the occurrence of gaze fixation does not necessarily mean that visual attention has occurred. Therefore, it was necessary to determine the occurrence of visual attention by setting the minimum time range for the degree of recognition according to the gaze time at which the gaze was fixed. Based on the previous research data shown in Table 1, 0.3 s was set as the minimum gaze time needed to acquire the visual characteristics in the range of cognition and perception of the object.

#### 2.2.2. Eye Tracking Analysis Method

The software provided with the eye tracking apparatus recorded all gaze movements as a path. Any point unrelated to color saliency and where the duration of the fixation was less than 0.1 s were excluded from the analysis [31,32]. Among the colors of the dental unit chair, the most viewed point, duration of fixation point(s), and the first fixation point were extracted (Table 2).

### 2.3. Color Analysis Based on Color Preference and Psychological Color Image

Eye tracking was used to analyze the gaze characteristics of the subjects, and a survey was conducted in parallel to compare the effects of color on user’s emotions. The color preference for the dental unit chair was examined by selecting 12 commonly used colors, and the general color preference of the subjects as well as the adjectives for colors was additionally investigated using color image analysis to identify any correlation. The same color values were used for the general color preference and the color preference for the dental unit chair, and we examined whether the general color preference matched the color preference for the dental unit chair considering the characteristics of the product. A total of 22 copies of the collected color–emotion questionnaire [16], which was scored based on the choice of color image adjectives, was used for the color analysis of the color preference and psychological color images using the Tobii analysis program.

#### Composition of the Survey

(1)General characteristics: gender and age(2)Color preference: select the general color preference and the color preference for a dental unit chair.(3)Psychological color image associated with the color preference for a dental unit chair: select the adjectives describing the psychological color image associated with the color preference for a dental unit chair.

In this study, based on the Color Image Scale developed by Kobayashi and the IRI Adjective Image Scale developed by the Research Institute for the Visual Language of Korea (Image Resource Institute), the IRI Hue & Tone 120 color system was used to extract data on the 12 colors commonly used for a dental unit chair for the survey. As for the color extraction method, computer mosaic analysis of the selected representative images was used to extract the color palette based on the color that occupied the widest area and the color that determined the impression (Table 3).

A total of 12 representative colors were extracted for the dental unit chair through this process (Table 4).

## 3. Results

### 3.1. Dominant Gaze Characteristics of All Subjects

In order to determine whether eye movements were affected by color preference, an analysis of the dominant gaze characteristics by color was performed. In Table 5, the colors that attracted the gaze were extracted by measuring the number of gazes and time. There was a total of 22 subjects, and in the correlation analysis of gaze characteristics and color preference, invalid data were excluded from the eye tracking results.

Data on the most dominantly observed colors during the 10 tests for each subject are shown. The first fixation point(s) in Table 5 refer to the color where the gaze stayed for the first time, and this was used to identify the place that attracted the gaze first in addition to the place where the gaze stayed for the longest time. In order to identify the first conscious gaze, data of 0.3 s or more, which was considered the conscious gaze time, were extracted and used for analysis.

Figure 4 shows the results of the gaze data analysis for the colors viewed by each subject. Based on the results for the most viewed point and first fixation point, the color on which the eyes of the subjects first stayed was pink (40.9%), and the color on which the subjects gazed the most was also pink (50%). The case where the most viewed point and the first fixation matched indicated that the color that attracted first attention was the same as the color the gaze stayed on the most, which accounted for 45.45%.

### 3.2. Color Preference Survey

In order to determine the relationship between the color preference and the color of the most viewed image, the gaze of the subjects was analyzed by color using the eye tracking results. Color preference was investigated by using a survey, and the color preference for the dental unit chair was examined by an additional survey as the general color preference might be different from the color preference for the dental unit chair, taking into account the characteristics of the specific product. Figure 5 shows the survey results.

Many of the subjects selected sky blue (28%) as their general color preference, and brown (16%) as their preferred color for a dental unit chair. The association between the color preference and the gaze was investigated using the dominant gaze data from the eye tracking results and the post survey results (before and after response). Figure 6 shows the results. There was no clear correlation between the most viewed point and the first fixation point, indicating that the first place that attracted the gaze was not necessarily the preferred color (Figure 6A). While some subjects showed consistent results in the selection of their general color preference and their color preference for the dental unit chair, no clear correlation was identified (Figure 6B). The faster and longer the first fixation was, the quicker the attention was drawn and the longer the attention was observed. However, as shown in Figure 6C–F, no distinct characteristics were observed. However, the fact that pink was the most viewed color in the eye tracking analysis while brown was selected as the most preferred color for the dental unit chair in the survey suggested that pink was not the most preferred color but the most eye-catching color.

### 3.3. Results of Color Analysis Based on Color Preference and Psychological Color Image

Figure 7 shows the results of analysis after creating the “IRI Adjective Image Scale” based on the results of the survey. The results were obtained by grouping the most preferred colors and adjectives selected on the “IRI adjective image scale” to determine the color preference for the dental unit chair as well as the image associated with the color. As for the adjective image scale, the top two groups with the highest number were separately grouped and displayed. The most preferred color for the dental unit chair in the survey was brown, and the subjects were asked to select multiple adjectives associated with the color without priority among the 12 representative adjectives used for the IRI adjective image scale, including <Pretty>, <Pure>, <Mild>, <Cheerful>, <Natural>, <Luxurious>, <Elegant>, < Peaceful>, <Dynamic>, <Modern>, <Noble>, and <Gentle>. Of the 12 adjectives, the most selected adjective was <Peaceful>, followed by <Natural> in second place, <Pretty> and <Elegant> in third place, and <Modern>, <Mild>, and <Gentle> in fourth place.

According to the “psychological color image analysis using the IRI color image scale” as shown in the Figure 7, <Natural> and <Peaceful> were selected as the most preferred adjective images. The emotions generated by the psychological color image in the two groups were as follows.

Natural: Plain, Intimate, Wholesome, DomesticPeaceful: Simple and Appealing, Peaceful, Gentle

## 4. Discussion

In regard to the eye tracking results, a longer average fixation indicated that the subject was interested and kept looking, and a faster and longer first fixation indicated that the object attracted the gaze of the subject more quickly and for longer. Many fixation counts indicated that gazes with interest and attention occurred multiple times. The eye tracking revealed that the same element induced different visual attention depending on the color. According to the survey, most of the subjects selected sky blue (28%) as their general color preference, and brown (16%) as their color preference for a dental unit chair. The cases where the general color preference did not match the color preference for the dental unit chair suggested that the color preference could depend on the characteristics of the product, which was the dental unit chair (Figure 5). Pink, attracted the attention of more than half of the subjects, and therefore, could be considered as one of the most eye-catching and preferred colors for the dental unit chair. However, contrary to our expectation that the eye tracking result would represent the color preference, the color preference and the eye tracking result did not match. When the first fixation point coincided with the most viewed point, it meant that the point attracted the gaze of the subject faster with interest and attention. This may have been due to the phenomenon of being attracted to the color that stands out the most amongst many colors or a sense of curiosity about a color that the subject has not experienced before as a color for a dental unit chair. Therefore, further investigations and studies are needed on color preference and attention.

In the psychological color image analysis results based on the color preference, the most dominant psychological color images associated with the color preference for the dental unit chair were <peaceful> and <natural>, regardless of the color preference results. This indicated that the psychological color image that came to mind was similar even if the preferred color of the dental unit chair varied according to individual preferences. Aside from the color of visual preference, the psychological color image represents the psychology of users who prefer and need the friendly, comfortable, and calm feelings that are induced by a <peaceful> and <natural> atmosphere.

In studies on the effect of color on emotion, behavior and response to color, brown has been reported to provide a “high level of comfort” and rational and mental stability [33].

The brown color selected by most of the subjects and the images associated with the preferred color, <peaceful> and <natural>, were in line with the results of previous studies reporting the preference for images such as affectionate, natural, emotional, warm, friendly, comfortable, tidy, simple, calm, neat, and mellow. This seemed to reflect the psychology of the users, who preferred and needed a feeling of comfort and tranquility from the color to help overcome the fear and anxiety experienced upon entering the dental clinic.

This suggests that that colors that provide a sense of peace and stability should be prioritized in creating a dental treatment environment, rather than the highly eye-catching color detected based on the eye tracking results.

In this study, the same element induced different types of visual attention depending on the color. In addition, the emotional reaction of the user for color image recognition was examined by the color preference and psychological color image surveys.

It is difficult to generalize the result of color preference based on the gaze analysis as a method for establishing a color scheme for a medical environment. Considering the color perception characteristics shown by dental hospital visitors for dental unit chairs, the results of this study could be used as basic data for establishing a color scheme for manufacturing dental unit chairs.

Based on the above results, this study presented the color preference and psychological color images based on the emotions of dental hospital users, and described the significance and limitations of the study.

## 5. Conclusions

The results of this study were obtained by understanding the relationship between color preference and eye movement and through comparing the eye tracking results with the color preference and associated psychological color image survey results. The results indicate that we can improve the patients’ satisfaction with the treatment space by considering visual components when designing the dental treatment environment and unit chair. They help to identify the visual factors that can improve the standard dental treatment environment. However, the visual tracking data may not be applicable to all standard dental treatment environments because we only tested a few patients. In further studies, we will consider additional samples and images to establish a positive emotional environment for dental treatment.

## Figures and Tables

**Figure 1 ijerph-18-07981-f001:**
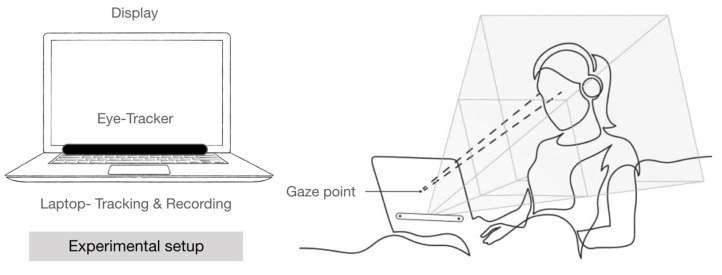
Experimental setup.

**Figure 2 ijerph-18-07981-f002:**
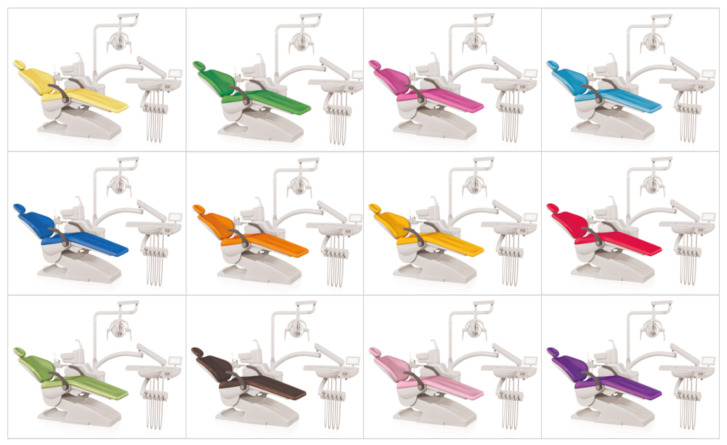
Color set of dental unit chair.

**Figure 3 ijerph-18-07981-f003:**
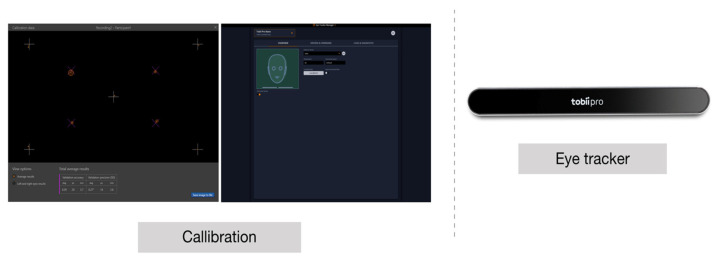
Calibrating and validating the eye tracker.

**Figure 4 ijerph-18-07981-f004:**
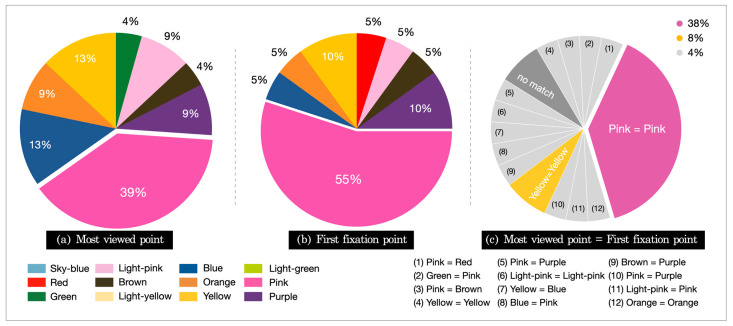
Gaze characteristics for the dental unit chair: (**a**) most viewed, (**b**) first fixation, (**c**) matched.

**Figure 5 ijerph-18-07981-f005:**
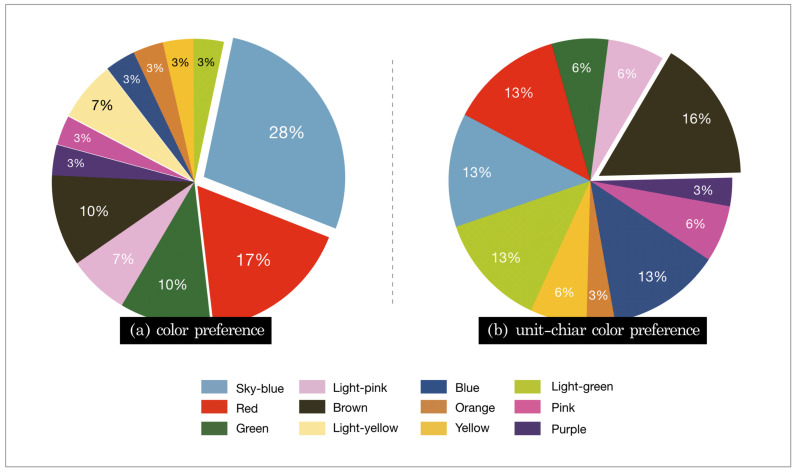
Color preference: (**a**) general and (**b**) dental unit chair.

**Figure 6 ijerph-18-07981-f006:**
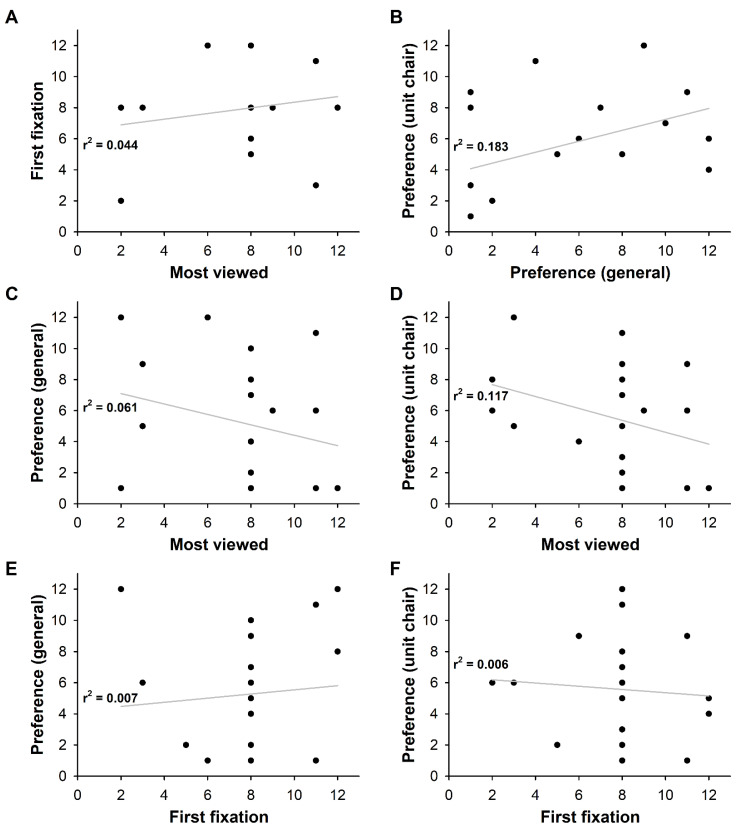
Correlation analysis for eye tracking and color preference. 1: sky blue, 2: light pink, 3: blue, 4: light green, 5: red, 6: brown, 7: orange, 8: pink, 9: green, 10: light yellow, 11: yellow, 12: purple. (**A**) Correlation between most viewed and the first fixation, (**B**) Correlation between color preference of unit chair and general color preference, (**C**) Correlation between most viewed and general color preference, (**D**) Correlation between most viewed and color preference of unit chair, (**E**) Correlation between the first fixation and general color preference, (**F**) Correlation between the first fixation and color preference of unit chair.

**Figure 7 ijerph-18-07981-f007:**
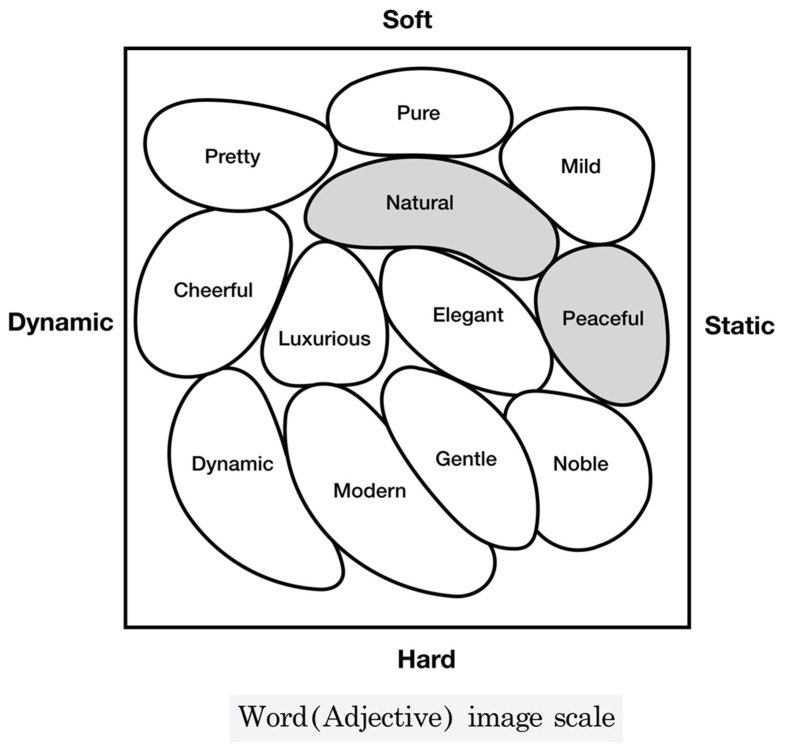
IRI word (adjective) image scale for dental unit chair.

**Table 1 ijerph-18-07981-t001:** Previous literature on gaze time.

Gaze Time	Definition	Previous Literature
0.2~0.25 s	Minimum time to focus on the object	Robert [25]
0.2 s	Minimum time to focus on the object	Gratzer [26]
0.25 s	Time of pre-saccadic attention shifts	Deubel [27]
0.25 s	Visual-fixed average time	Lovegrove [28]
0.05 s	Minimum time to visual continuity	Berger [29]
0.02~0.04 s	Visual-fixed minimum time	David [30]

**Table 2 ijerph-18-07981-t002:** Eye tracking analysis elements and definitions.

Analysis Method	Definition
1. First fixation point	User’s first gaze point
2. Duration of first fixation points	Time of user’s first gaze point
3. Most viewed point	Most repeated user’s gaze point
4. Fixation count of most viewed point	Count of most repeated user’s gaze point
5. Total duration of fixations	Accumulated time of user’s gaze point (fixed time > 0.1 s)

**Table 3 ijerph-18-07981-t003:** Color data extraction method.

Selected Image	Mosaic Analysis	Color Palette Extraction	Extracted Data
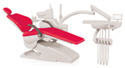	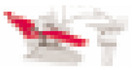	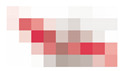	
RGB	3,185,378
Hex Code	DA354E

**Table 4 ijerph-18-07981-t004:** Data for 12 extracted colors frequently used for a dental unit chair.

Dental Unit Chair	Extracted Color	Color Value	Dental Unit Chair	Extracted Color	Color Value
RGB	Hex Code	RGB	Hex Code
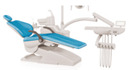		110,196,230	6EC4E6	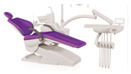		13,277,149	844D95
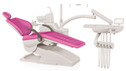		218,115,156	DA739C	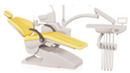	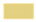	249,227,140	F9E38C
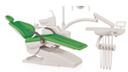	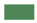	5,014,081	328c51	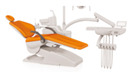	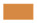	2,491,420	F98E00
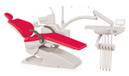	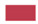	3,185,378	DA354E	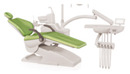	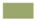	16,621,595	A6D75F
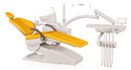	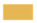	24,419,162	F4BF3E	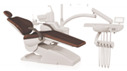	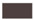	604,848	3C3030
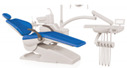		1,482,140	0E528C	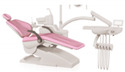	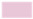	242,192,215	F2C0D7

Color analysis was conducted based on the IRI adjective image scale.

**Table 5 ijerph-18-07981-t005:** Average gaze data for gaze color by subject.

Tester	Recording Time (s)	Most Viewed	Total Duration of Fixation (s)/Avg/Max	First Fixation	Fixation Count of Most Viewed	Duration of First Fixation
1	4.90	Pink	4.96/1.24/2.08	Pink	5	0.38
2	4.93	Pink	3.53/1.675/2/1	Red	6	0.34
3	4.87	Green	2.34/1.17/1.57	Pink	2	0.31
4	4.84	Pink	1.95/0.975/1.1	Brown	4	0.52
5	4.89	Yellow	2.94/0.98/1.32	Yellow	6	0.31
6	4.86	Brown	3.95/1.975/2.37	Purple	5	0.95
7	4.79	Pink	4.46/1.115/1.55	Purple	3	0.54
8	4.89	Light-pink	2.7/1.35/1.42	Light-pink	4	0.31
9	4.92	Yellow	1.74/0.87/0.97	Blue	3	0.37
10	4.93	Purple	4.25/1.417/2.53	Pink	2	0.73
11	4.79	Blue	1.92/0.96/1.02	-	4	-
12	4.79	Blue	2.93/0.977/1.31	Pink	5	0.36
13	4.83	Pink	5.11/1.7/2.19	Pink	6	0.42
14	4.84	Light-pink	5.27/1.76/2.4	Pink	6	0.62
15	4.78	Orange	3.56/1.78/2.86	Orange	9	0.46
16	4.75	Pink	6.02/1.50/1.75	Pink	3	0.43
17	4.76	Yellow	2.86/0.95/1.74	Yellow	3	0.3
18	4.73	Pink	4.01/1.34/1.88	Pink	2	2.12
19	4.80	Pink	6.71/1.68/2.4	Pink	2	1.34
20	4.72	Pink	3.67/1.84/2.19	Pink	6	0.5
21	4.87	Purple	3.14/1.05/1.32	-	5	-
22	4.99	Blue	3.38/1.69/2.55	Pink	3	0.37

## Data Availability

The data presented in this study are available on request from the corresponding author.

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
