# Peer review of "Assessment of Color Perception and Preference with Eye-Tracking Analysis in a Dental Treatment Environment"

_ijerph, 2021, doi:10.3390/ijerph18157981_

Round 1

Reviewer 1 Report

Please open the following file to access the revision

Author Response

POINT-BY-POINT RESPONSE FORM

Assessment of color perception and preference with eye-tracking analysis on dental treatment environment

We would like to thank the reviewer for careful and thorough reading of this manuscript and for the thoughtful comments and constructive suggestions, which help to improve the quality of this manuscript. Our response follows.

#Reviewer1

Comments and suggestions

The present article is of a great interest for this journal, although there are some criticalties that should be addressed. First of all, extensive English revision should be carried out in order to make the manuscript more fluent and understandable. Therefore, it is suggested to review the whole manuscript by a native English speaker. Moreover, it is the reviewer opinion that the whole abstract should be rewritten. It is indeed unclear and it should be reorganized in order to make it more appealing to the reader.

Response: Thank you for the reviewer’s insightful comment. We agree and carefully revised the level of English in this paper. We consider the issues mentioned result from wrong expressions that are related to errors in English and grammar. As you have suggested, we revised and rewritten the abstract.

- Line 17-31: Nowadays, people investigate the advanced service to develop medical environments for the patients. Especially, the dental hospital has been considered uncomfortable and afraid of the space, it has to change so people can easily visit and feel more relaxed. However, only a few systematic studies reported a demand for building a comfortable space. this study aims to inves-tigate gaze characteristics based on the color preference survey of the dental unit chair, which has the most influence on spatial perception in the dental treatment environment, using the eye tracking technique for color. As a result of this study, there was a tendency that the color percep-tion by eye tracking and the color preference by survey did not match. The color most viewed by a majority of subjects was the pink color attracting a high level of attention, regardless of their personal preference. In addition, as for the psychological color images associated with col-or preference, the subjects tended to prefer the images such as warmth, friendliness, and calm-ness. This appeared to reflect the psychology of the subjects who wished to replace the anxious feelings such as fear when coming to the dental hospital with comfort and tranquility. Therefore, the colors that can give comfort and tranquility to patients should be considered first as visual elements (brown color etc.) in creating a dental treatment environment.

Please check carefully if the Keywords selected are present in MeSH database, this would greatly improve the article visibility amongst the main search engines.

Response: Thank you for your comments. We modified it by referring to the keyword in the Mesh database.

- Line 32: color, psychological color, eye tracking, perception, preference, dental treatment environment

,

Now there will be some point-by-point suggestions to the authors:

INTRODUCTION: This section should give the reader a proper background regarding the topic addressed during the rest of the manuscript

  1. The passage from line 55 to 62 is unclear, please revise it

Response: Thank you for your comments. We revised the sentences on Line 55 to 62.

- Line 53-59: Many patients feel uncomfortable about pain and hospital when they meet the dentist [8-11]. It suggests that the hospital workers, including the doctors they care about, create a psychologically satisfied environment not only for patients but also themselves when de-signing the dental space. It helps to provide relaxation and comfortable space to patients that design the rules of light and color in interior/exterior space [12,13]. The dental unit chair, directly receives treatment to the patients, is the most effective space and to design color can be helpful to strong visual expression in dental care space [14,15].

  1. Line 83-86, the authors say that other studies have found some correlation… it is the revisor idea that only 3 studies are too few to affirm something like this, please add more citations.

Response: Thank you for your comments. We added more references.

- Line 80-83: Studies to find the correlation between color and humans reported that the infor-mation given by color was closely associated with emotion through analysis of people's preference for color, the effect of color on emotions, and behavior and response to color [18-24].

  1. MANAV, Banu. Color‐emotion associations and color preferences: A case study for residences. Color Research & Application: Endorsed by Inter‐Society Color Council, The Colour Group (Great Britain), Canadian Society for Color, Color Science Association of Japan, Dutch Society for the Study of Color, The Swedish Colour Centre Foundation, Colour Society of Australia, Centre Français de la Couleur, 2007, 32.2: 144-150.
  2. Schloss, K. B., & Palmer, S. E. (2009). An ecological valence theory of human color preferences. Journal of Vision, 9(8), 358-358.
  3. Lee, C. J., & Andrade, E. (2010). The effect of emotion on color preferences. ACR North American Advances.
  4. Hurlbert, A., & Owen, A. (2015). Biological, cultural, and developmental influences on color preferences.

The Discussion and the Conclusion are listed in the same paragraph, please carefully read the instructions to the authors and rearrange this section. These two sections should be separated because they have different purposes.

Response: Thank you for your comments. We have distinguished the Discussion and Conclusion.

- Line 243-292: In the eye tracking result, a longer average fixation indicated that the subject was interested and kept looking, and a faster and longer first fixation indicated that the object attracted the gaze of the subject faster and longer. Many fixation counts indicated that gazes with interest and attention occurred multiple times. The eye tracking revealed that the same element induced different visual attention depending on the color. As a result of the survey, most of the subjects selected sky blue (28%) as their general color preference, and brown (16%) as their color preference for a dental unit chair. The cases where the general color preference did not match the color preference for the dental unit chair suggested the color preference could depend on the characteristics of the product, which was the dental unit chair (Figure 5). Pink, which attracted the attention of more than half of the subjects, could be considered as one of the most eye-catching colors for establishing the color of the dental unit chair. However, unlike the expectation that the eye tracking result would rep-resent the color preference, the color preference and the eye tracking result did not match. When the first fixation point coincided with the most view point, it meant that the point attracted the gaze of the subject faster with interest and attention. This may have been due to the phenomenon of being attracted to the color that stood out the most amongst many colors or the curiosity for the color that a subject had never experienced before as the color for a dental unit chair. Therefore, further investigations and studies are needed on color preference and attention.

In the psychological color image analysis results based on the color preference, the most dominant psychological color images associated with the color preference for the dental unit chair were <peaceful> and <natural>, regardless of the color preference results. This indicated that the psychological color image that came to mind was similar even if the preferred color of the dental unit chair varied according to individual preferences. Aside from the color of visual preference, the psychological color image represented the psychology of the users who preferred and needed the friendly, comfortable, and calm feelings from the <peaceful> and <natural> atmosphere.

In studies on the effect of color on emotion, behavior and response to color, the brown color has been reported to provide a "high level of comfort" with rational and mental stability [33].

The brown color selected by most of the subjects and the images associated the preferred color, <peaceful> and <natural>, were in line with the results of previous studies reporting the preference for the images such as affectionate, natural, emotional, warm, friendly, comfortable, tidy, simple, calm, neat, and mellow. This seemed to reflect the psychology of the users who preferred and needed a feeling of comfort and tranquility from the color to overcome the fear and anxiety experienced upon entering the dental clinic.

This suggested that that the color giving a sense of peace and stability should be prioritized in creating a dental treatment environment, rather than the highly eye-catching color based on the eye tracking result.

In this study, the same element induced different visual attention depending on the color. In addition, the emotional reaction of the user for color image recognition was examined by the color preference and psychological color image surveys.

It would be difficult to generalize the result of color preference based on the gaze analysis as a method for establishing a color scheme for a medical environment. Considering the color perception characteristics shown in the perception process of dental unit chair by dental hospital visitors, the result of this study can be used as basic data for establishing a color scheme for manufacturing dental unit chairs.

Based on the above results, this study presented the color preference and psychological color images based on the emotions of dental hospital users, and described the significance and limitations of the study.

- Line 294-303: The results of this study were obtained by understanding the relationship between color preference and eye movement through comparing the eye tracking results with the color preference and associated psychological color image survey results. This result indicates we improved the patients satisfied with the relaxation space to visualizing components when designing the dental treatment environment and unit chair. It helps the visualizing factors to make the standard dental treatment environment through our results. However, this visual tracking data does not indicate the standard dental treatment environments because we tested the few patients. In further study, it will be considered that additional study about samples and images to establish the emotional environments of the dental treatment. 

Please carefully read the instructions to the authors, the bibliography is not uniform, it is crucial to revise it properly. The manuscript overall presents an important issue in the dentistry field, and for this reason is very important to carefully revise the whole article in order to be published.

Response: Thank you for your comments. We checked again the references and corrected it.

- Line 319-384:

  1. Huisman, E.R.C.M.; Morales, E.; van Hoof, J.; Kort, H.S.M. Healing Environment: A Review of the Impact of Physical Environmental Factors on Users. Environ. 2012, 58, 70-80. https://doi.org/10.1016/j.buildenv.2012.06.016
  2. Gedam, K.; Katre, A. Scenario-Based Assessment of Children and Parents Preferences towards a Paediatric Dental Setup – An Observational Study. IOSR Dent. Med. Sci. 2018, 17, 32-42.
  3. Rafeeq, D.A.; Mustafa, F.A. Evidence-Based Design: The Role of Inpatient Typology in Creating Healing Environment, Hospitals in Erbil City as a Case Study. Ain Shams Eng. J. 2021, 12, 1073-1087. https://doi.org/10.1016/j.asej.2020.06.014
  4. Aripin, S. Healing Architecture: Daylight in hospital design. Proceedings of the Conference on Sustainable Building South-East Asia, Johor, Malaysia, 5-7 Nov 2007
  5. Woo, J.-C.; Lin, Y.-L. Kids’ Perceptions toward Children’s Ward Healing Environments: A Case Study of Taiwan University Children’s Hospital. Healthc. Eng. 2016, 2016, 8184653. https://doi.org/10.1155/2016/8184653.
  6. Umamaheshwari, N.; Asokan, S.; Kumaran, T. Child Friendly Colors in a Pediatric Dental Practice. Indian Soc. Pedod. Prev. Dent. 2013, 31, 225-228. https://doi.org/10.4103/0970-4388.121817.
  7. Sojan, M., Thakur, S., Chauhan, D. Fundamentals in Paediatric Dental Clinic Set – up: A Comprehensive Review. J. Res. Rep. Dent. 2021, 4, 1-6. https://www.journalijrrd.com/index.php/IJRRD/article/view/30135
  8. Folayan, M.O.; Adekoya-Sofowora, C.A.; D. Otuyemi, O.; Ufomata, D. Parental Anxiety as a Possible Predisposing Factor to Child Dental Anxiety in Patients Seen in a Suburban Dental Hospital in Nigeria. J. Paediatr. Dent. 2002, 12, 255-259. https://doi.org/10.1046/j.1365-263X.2002.00367.x.
  9. Yamada, M.K.M.; Tanabe, Y.; Sano, T.; Noda, T. Cooperation during Dental Treatment: The Children’s Fear Survey Schedule in Japanese Children. J. Paediatr. Dent. 2002, 12, 404-409. https://doi.org/10.1046/j.1365-263X.2002.00399.x.
  10. Guo, B., Liu, J., Yang, F., Xie, S.J., Que, K.H., Zhang, Q. The prevalence of dental anxiety in the aged people. J. stomatol. 2007, 34, 162-164.
  11. Townend, E.; Dimigen, G.; Fung, D. A Clinical Study of Child Dental Anxiety. Res. Ther. 2000, 38, 31-46. https://doi.org/ 10.1016/S0005-7967(98)00205-8.
  12. Mohammad, K. A Standard Pediatric Dental Clinic. App. Dent. Oral Health. 2018, 3. http://dx.doi.org/10.32474/MADOHC.2018.03.000152
  13. American Dental Association. Building or Refreshing Your Dental Practice: A Guide to Dental Office Design; American Dental Association: IL, USA, 2017; pp. 1–312.
  14. Kamali, N.J.; Abbas, M.Y. Healing Environment: Enhancing Nurses’ Performance through Proper Lighting Design. Procedia - Social Behav. Sci. 2012, 35, 205-212. https://doi.org/10.1016/j.sbspro.2012.02.080
  15. Aripin, S. Healing architecture: A study on the physical aspects of healing environment in hospital design. In Proceedings of the 40th Annual Conference of the Architectural Science Association (ANZAScA), Adelaide, South Australia, 22–25 November 2006; Australia Shannon, S., Soebarto, V., Williamson, T., Eds.; Architectural Science Association (ASA): Adelaide, Australia, 2006; pp. 342–349.
  16. Kobayashi, S. The Aim and Method of the Color Image Scale. Res. Appl. 1981, 6, 93-107. https://doi.org/10.1002/col.5080060210.
  17. Baek, C.-H.; Park, S.-O.; Kim, H.-S. The Analysis of Emotion Adjective for LED Light Colors by Using Kobayashi Scale and I.R.I Scale. Korean Ins. Illuminat. Electric. Install. Eng. 2011, 25, 1-13. https://doi.org/10.5207/JIEIE.2011.25.10.001.
  18. Guilford, J.P. Smith, P.C. A system of color-preferences. J. psychol. 1959, 72, 487–502. https://doi.org/10.2307/1419491
  19. Wexner, L.B. The Degree to Which Colors (Hues) Are Associated with Mood-Tones. App. Psychol. 1954, 38, 432-435. https://doi.org/10.1037/h0062181.
  20. Adams, F.M.; Osgood, C.E. A Cross-Cultural Study of the Affective Meanings of Color. Cr. Cul. Psychol. 1973, 4, 135-156. https://doi.org/10.1177/002202217300400201.
  21. Manav, B. Color-emotion associations and color preferences: A case study for residences. Res. App. 2007, 32, 144-150. https://doi.org/10.1002/col.20294
  22. Palmer, S.E.; Schloss, K.B. An ecological valence theory of human color preference.  Natl. Acad. Sci. U.S.A.2010, 107, 8877–8882. https://doi.org/10.1073/pnas.0906172107
  23. Lee, C.J.; Andrade, E. The effect of emotion on color preferences. ACR Nor. Am. Adv. 2010, 37, 846-847. https://www.acrwebsite.org/volumes/15490/volumes/v37/NA-37
  24. Hurlbert, A.; Owen, A. Biological, cultural, and developmental influences on color preferences. In Handbook of color psychology; A. J. Elliot, M. D. Fairchild, A. Franklin; Cambridge University Press: Cambridge, England, 2015; pp. 454–477.
  25. Robert, L.S. Cognition and the visual arts; The MIT Press: MA, USA, 1996
  26. Gratzer, M.A., McDowell, R.D. Adaptation of an eye movement recorder to esthetic environmental mensuration. Research Report. Storrs Agricultural Experiment Station, University of Connecticut. 1971, 36, 3-29.
  27. Deubel, H. The Time Course of Presaccadic Attention Shifts. Res. 2008, 72, 630-640. https://doi.org/10.1007/s00426-008-0165-3.
  28. Lovegrove, W.J.; Garzia, R.P.; Nicholson, S.B. Experimental evidence for a transient system deficit in specific reading disability. Am. Optom. Assoc. 1990, 61, 137-146.
  29. Berger, A.A. Seeing Is Believing: An Introduction to Visual Communication. Aes. Art Crit. 1991, 49, 101-102. https://doi.org/10.2307/431664
  30. Grisham, D. Developing Ocular Motor and Visual Perceptual Skills: An Activity Workbook. Vis. Sci. 2005, 82, 941-942. https://doi.org/10.1097/01.opx.0000188478.09025.7f
  31. Itti, L.; Koch, C.; Niebur, E. A Model of Saliency-Based Visual Attention for Rapid Scene Analysis. IEEE Trans. Pat. Anal. Mach. Int. 1998, 20, 1254-1259. https://doi.org/1109/34.730558
  32. Zhao, Q.; Koch, C. Learning a Saliency Map Using Fixated Locations in Natural Scenes. Vis. 2011, 11, 9-9. https://doi.org/10.1167/11.3.9
  33. Murray, D.C.; Deabler, H.L. Colors and Mood-Tones. Appl. Psychol. 1957, 41, 279-283. https://doi.org/10.1037/h0041425.

Reviewer 2 Report

The paper points the attention on the never enough studied importance of a confortable environment where to treat our patients. Especially in dentistry, where the dentist is associated to the unpleasant idea and sound of a painful drill, a peaceful place is highly desirable. The study in general is well conducted and, in its semplicity, it is coherent and effective showing how the desired idea often is different from the most eye-catching or pleasable one. Written in good English language but there are few typing mistakes.

Author Response

POINT-BY-POINT RESPONSE FORM

Assessment of color perception and preference with eye-tracking analysis on dental treatment environment

#Reviewer2

Comments and suggestions

The paper points the attention on the never enough studied importance of a confortable environment where to treat our patients. Especially in dentistry, where the dentist is associated to the unpleasant idea and sound of a painful drill, a peaceful place is highly desirable. The study in general is well conducted and, in its semplicity, it is coherent and effective showing how the desired idea often is different from the most eye-catching or pleasable one. Written in good English language but there are few typing mistakes.

Response: Thank you for appreciating our study. We checked the manuscript again and revised the overall English.